# Valorizing *Carasau* Bread Residue Through Sourdough Fermentation: From Bread Waste to Bread Taste

**DOI:** 10.3390/microorganisms13081745

**Published:** 2025-07-25

**Authors:** Simonetta Fois, Valentina Tolu, Vanna Sanna, Antonio Loddo, Manuela Sanna, Piero Pasqualino Piu, Daniela Piras, Tonina Roggio, Pasquale Catzeddu

**Affiliations:** 1Porto Conte Ricerche, Loc. Tramariglio, 07041 Alghero, Italy; fois@portocontericerche.it (S.F.); tolu@portocontericerche.it (V.T.); sannav@portocontericerche.it (V.S.); sanna@portocontericerche.it (M.S.); piu@portocontericerche.it (P.P.P.); pirasd@portocontericerche.it (D.P.); roggio@portocontericerche.it (T.R.); 2MFM Sunalle, Via Ogliastra 10, 08023 Fonni, Italy; antonio.loddo@sunalle.it

**Keywords:** *Lactiplantibacillus plantarum*, *Saccharomyces cerevisiae*, *Wickerhamomyces anomalus*, semolina, wheat middlings, carasau bread, circular economy

## Abstract

Surplus bread accounts for a significant proportion of food waste in many countries. The focus of this study was twofold: firstly, to investigate the use of carasau bread residue as a sourdough substrate, and secondly, to reuse this sourdough into a new carasau baking process. Selected lactic acid bacteria (*Lactiplantibacillus plantarum*) and yeast strains (*Saccharomyces cerevisiae* and *Wickerhamomyces anomalus*) were used to inoculate three substrates: bread residue (S1), bread residue supplemented with durum wheat middlings (S2), and semolina (S3). Sourdoughs were refreshed for five days by backslopping, and microbiological and physicochemical analyses were performed. Results indicated that incorporating wheat middlings into bread residue enhanced microbial performance, as evidence by a decrease in pH from 6.0 to around 4.5 compared to using bread residue alone as a substrate. Carasau bread produced with the sourdough derived from bread residue and wheat middlings exhibited comparable physicochemical properties to commercial baker’s yeast carasau bread, but had better sensory properties, scoring a mean acceptability of 7.0 versus 6.0 for baker’s yeast bread. These results show that bread residue supplemented with wheat middlings can serve as a sourdough substrate, allowing its reuse in the baking process to produce high-quality carasau bread and promote the circular economy.

## 1. Introduction

Every year, about 900,000 tons of bread is wasted worldwide across the supply chain by manufacturers and consumers, and in many countries bread is a primary contributor to food waste [1]. The transformation of bread into waste can have several origins, with staling and spoiling the leading causes, both responsible for a reduction in shelf life. Additionally, dropped, scraped, or surplus bread resulting from bread manufacturing and the supply chain also contributes [2]. As a result, bakeries endure considerable annual economic losses due to this massive waste of unsold bread, much of which is returned to manufacturers. Consumers also contribute to this waste by discarding leftover bread at home. Beyond the wasted bread itself, significant resources like water and energy are also dissipated, along with those involved in producing, transporting, and manufacturing the raw materials [3]. Therefore, reducing or recycling bread waste is a crucial challenge for preserving economic value and bioresources. Significant efforts have been made to develop technologies for bread waste valorization. For instance, Cacace et al. [4] confirmed bread waste’s potential as an organic soil amendment, showing benefits in alkaline soils, but not in acidic ones. Other alternative applications that have been explored include its use in textiles, in the production of graphene, and in the extrusion process, as well as in beer brewing [5].

Using bread residue as a food ingredient seems like the most logical way to reduce waste. However, as Gomez and Martinez [6] recently concluded in their review, bread residue should only be incorporated in limited quantities into bread and other baked goods. In bread-making, the presence of denatured proteins and the inability to form a gluten network reduce the bread’s volume. For baked goods, the increased water retention of gelatinized starch negatively affects their consistency. Furthermore, these authors highlighted the importance of the safety requirements for this ingredient, including risks of microbial contamination and mycotoxin.

Currently, bread waste serves as a cost-effective and readily available feedstock for microbial fermentation [5,7]. To enhance microbial growth, enzymes are often added prior to the fermentation process to the bread substrate to hydrolyze starch and/or proteins. Subsequently, selected microorganisms are used to produce industrial valuable chemicals, including ethanol, lactic acid, and succinic acid. Supplementation of bread waste with inorganic salts, such as ammonium sulfate and potassium phosphate, has been shown to enhance microbial growth. Asghar et al. [8] demonstrated that addition of MgSO_4_·7H_2_O, CaCl_2_·2H_2_O and KH_2_PO_4_ improved α-amylase production through microbial fermentation. Enhancement of yeast biomass production in bread medium containing protease and salts (KH_2_PO_4_ and NH_4_SO_4_) was demonstrated by Benabda et al. [9]. Furthermore, the potential of bread waste for cultivating food industry starters and as a substrate for sourdough production has also been explored [10,11,12]. In the abovementioned studies, the bread substrate was supplemented with enzymes and/or inorganic salts to enhance carbon and nitrogen availability for microbial growth. However, these supplements increase the cost of fermentation process, undermining the economic advantage of using bread waste as a substrate.

In this context, we aimed to develop an approach that valorizes and recycles bread residue by using an inexpensive and easy-to-use supplement to produce new bread, thereby offering bakeries a sustainable and economically advantageous solution without any compromise on the final product’s quality.

*Carasau* is a type of flat bread, thin and crispy, that is baked twice at around 500 °C [13]. During the first baking, leavened dough sheets (about 2 mm thick) puff into a spherical shape due to CO_2_ expansion and rapid moisture evaporation, and then deflate to form a two-layer structure. Before the second baking stage, which is designed to toast bread, the two layers are split, either manually or mechanically, thus generating about 10% of residues, which are typically repurposed as animal feed.

This study investigated the use of carasau bread-making residues as a substrate for sourdough fermentation, aiming to reuse this sourdough in new carasau bread production, aligning with the modern principles of a circular economy. Selected strains of lactic acid bacteria (LAB) and yeast were used as starters for sourdough production. Fermentations were carried out on three different substrates: bread residue (S1), bread residue supplemented with durum wheat middlings (S2), and a semolina control (S3). The decision to include wheat middlings stemmed from prior experimentation, their cost-effectiveness, and their market availability. The experimental sourdoughs were refreshed for a week using the backslopping method, followed by thorough chemical, physical, and microbiological analyses. The study further assessed the leavening properties of sourdough made from bread residue and wheat middlings in bread dough. Finally, sourdough was used as a leavening agent to make carasau bread, and physicochemical and sensory properties and consumer acceptance of the bread were evaluated.

## 2. Materials and Methods

### 2.1. Ingredients and Sourdough Preparation

Treatment involved three different substrates (S1, S2, and S3), which were used in the production of sourdough. Two replicates of each treatment were carried out. Substrates were prepared as follows.

S1: Bread residue and water were mixed at a ratio of 1:2.5. This ratio was determined based on a calculated water binding capacity [14], which was 2.9 g/g of dry weight (d.w.).

S2: A blend of bread residue and wheat middlings (ratio 9:1) was mixed with water, maintaining a ratio of 1:2.5.

S3: Semolina and water were mixed at a ratio of 1:1 (resulting in a dough yield of 200) and used as control.

Bread residue was prepared as follows. Residue of carasau bread made from remilled semolina was dried at 130 °C for 20 min to reduce the moisture content from 20% to approximately 10%. After grinding into <2 mm crumbs, the bread residue was vacuum-packed and stored at 10 °C until use. Its protein content was 12.34% on d.w.

The durum wheat middlings were kindly provided by the F.lli Brundu mill (Macomer, Italy). The moisture (8.85%), lipid (6.1% d.w.), total dietary fiber (48.01% d.w.), ash (5.01% d.w.), and protein content (15.87% d.w.) was determined.

Semolina used for sourdough fermentation was kindly provided by Molino Simec (Oristano, Italy).

The microorganisms used for sourdough preparation belonged to the Microbial Culture Collection of Porto Conte Ricerche. Strains were chosen because of their beneficial metabolic properties, which are known to improve both the sourdough fermentation and the final quality of bread [15]. *L. plantarum* strain PCC2397, an LAB species considered very competitive [16], was selected for its amylolytic activity, acidifying properties, and ability to produce exopolysaccharides. *S. cerevisiae* strain PCC1618 was selected because of its high fermentation capacity. Among the non-conventional yeasts, the *W. anomalus* strain PCC1629 exhibited good resistance to acidic conditions and notable proteolytic activity. Additionally, this strain had the ability to impart a characteristic aromatic profile to the sourdough, which was preliminary evaluated through sensory analysis by trained judges.

To prepare the sourdough, the LAB strain was pre-cultured on de Man, Rogosa and Sharpe (MRS) broth medium at 28 °C. After 24 h, it was inoculated into 50 mL of fresh medium. The yeast strains were pre-cultured in YPG broth (containing 10.0 g/L yeast extract, 10.0 g/L bacteriological peptone, 20.0 g/L glucose) at 28 °C for 24 h under stirring conditions. Then, each strain was inoculated into 100 mL of fresh medium. After 24 h of incubation, the microbial cultures were centrifuged at 1298× *g* for 15 min at 4 °C. The harvested cells were resuspended in 5 mL of sterile physiological solution (0.9% NaCl, *w*/*v*) and used to inoculate the substrates—S1, S2, and S3.

The sourdoughs were managed using an AFTL5 bioreactor (SITEP S.r.l., Voghiera, Italy). The fermentation step was carried out at 25 ± 1.5 °C for 8 h based on previous experiments. After that, the substrate was refrigerated at 5 ± 1.5 °C. Sourdough refreshment was performed daily from Monday to Friday using the backslopping method. For S1 and S2, the mixture of sourdough, substrate, and water was combined in a ratio of 1:1:2.5. For S3, this ratio was 1:1:1. After fermentation, the fifth refreshment (R5) was refrigerated over the weekend and analyzed on Monday. Previous experiments showed that starting a semolina sourdough and refreshing it daily for five days consistently resulted in stable pH levels and microbial counts. Total titratable acidity (TTA), pH, and plate count analyses were performed on the refrigerated sourdoughs. Samples were taken 24 h after each refreshment.

To determine the viabilities of both LAB and yeasts in refrigerated sourdough samples, plate count analysis was performed in duplicate. Ten grams of sample were mixed with 90 mL of sterile peptone solution (0.1% peptone, *w*/*v*) and homogenized with a Stomacher Lab blender 80 (PBI, Milan, Italia). Serial dilutions were then prepared and plated onto MRS agar medium and on WL nutrient agar medium containing chloramphenicol for LAB and yeast, respectively. MRS plates were incubated at 30 °C for 48 h under anaerobiosis (Anaerogen System, Oxoid). WL nutrient agar plates were incubated at 30 °C for 48 h under aerobiosis. The two yeast species were differentiated on WL nutrient agar medium based on distinct colony morphology and color. *S. cerevisiae* colonies appeared blue with an off-white center, while *W. anomalus* colonies appeared white (Appendix A). All culture media and ingredients were purchased from Oxoid (Basingstoke, Hampshire, UK).

### 2.2. Chemical Analyses

Moisture and ash content was determined using a Thermostep thermogravimetric analyzer (Eltra GmbH, Haan, Germany). Samples were heated at 130 °C for moisture until a stable weight and at 580 °C for ash. Protein content (Nx5.7) was determined according to the AACC combustion method 46–30 [17] using a Rapid N Cube analyzer (Elementar Analysensysteme GmbH, Langenselbold, Germany). Total dietary fiber content was determined using a total dietary fiber assay kit (Megazyme, Wicklow, Ireland). TTA and pH were measured using an automatic titrator (Crison, Hach Lange, Barcelona, Spain) after homogenization of 10 g of sample in 90 mL of distilled water. After 30 min of gentle stirring, the samples were titrated to a pH of 8.5 using 0.1 N NaOH. The TTA value was then recorded as milliliters of NaOH per 10 g of sample. All analyses were performed in triplicate.

Carbohydrates, organic acids, and ethanol were determined as follows. An aliquot of 5.0 g of sample, sourdough, or bread was dispersed in distillated water (45 mL) and magnetically stirred at room temperature for 30 min. The dispersion was then centrifuged at 12,000× *g* for 10 min at 20 °C and the supernatant filtered through a 0.45 μm PTFE syringe filter. All collected extracts were stored at −80 °C until analysis. All samples were prepared and analyzed in duplicate. The results are presented as a percentage of content (% *w*/*w*). The analyses were performed using the Enzytec^TM^ kits from R-Biopharm (R-Biopharm AG, Darmstadt, Germany) following the manufacturer’s instructions on the iCubio i-Magic M9 (Origlia S.r.L, Cornaredo, Italy). The enzymatic reactions were fully automated, and the absorbance of NADH produced, which is proportional to concentration of each original metabolite, was measured at 340 nm.

The D-glucose concentration (g/L) in the samples was quantified using an Enzytec™ liquid D-glucose kit (E8140) accepted as an AOAC official method of analysis℠ [18].

The D-fructose content (g/L) was determined using an Enzytec™ liquid D-glucose/D-fructose kit (E8160), accepted as an AOAC Official Method of Analysis [18].

The total concentration of maltose (g/L) was measured using an Enzytec™ liquid maltose/sucrose/D-glucose kit (E8170). To determine the real maltose concentration, the sum of sucrose and free D-glucose in the sample had previously been quantified using an Enzytec™ liquid sucrose/D-glucose kit (E8180), accepted as an AOAC official method of analysis [19]. The real maltose concentration was calculated as follows:Cmaltose (g/L) = Ctotal maltose (E8170) − 0.5 × Ctotal sucrose (E8180).

The acetic acid content (g/L) was quantified using an EnzytecTM liquid acetic acid kit (E8226), accepted as an AOAC official method of analysis [20].

The D-/L-lactic acid concentration (g/L) was determined using an EnzytecTM liquid D-/L-lactic acid kit (E8240), accepted as an AOAC official method of analysis [21].

Total concentration of ethanol (g/L) was measured using an EnzytecTM liquid ethanol kit (E8340), accepted as an AOAC official method of analysis [22].

### 2.3. Determination of Sourdough Consistency

To study the effect of microbial fermentation on sourdough consistency, a “back-extrusion rig test” was conducted using a TA.XTplus texture analyzer (Stable Microsystems, Godalming, UK) equipped with a 5 kg load cell and probe P/35 cylinder aluminum using Exponent Stable Micro Systems (v6.1.16) software. The following parameters were used. pre-test, test, and post-test speed: 2 mm/s; trigger type: auto force; trigger force: 0.029 N; distance: 18 mm; break mode: off; stop plot at: start position. The probe performed a compression test, extruding the product up and around the cylinder’s edge. Each test was carried out in a 150 mL container with 100 g of sample, which was brought to 22 °C before analysis. Two determinations were made: the first was performed on sourdough before fermentation (T0), the second after the fourth refreshment (R4). Analysis was performed in triplicate. The area of the curve up to the maximum force was taken as a measurement of consistency (newtons per second, N s). The maximum force in the negative region of the graph indicated cohesiveness (N). The reduction in consistency and cohesiveness at R4 was calculated using the formula [(T0-R4)/T0 × 100].

### 2.4. Leavening Activity of Sourdough

The leavening ability of sourdoughs was evaluated in doughs inoculated with 5%, 10%, and 20% of sourdough. The same remilled semolina used for the bread-making trials, kindly provided by Il Vecchio Forno-Sunalle bakery (Fonni, Italy), was used for this test. Dough hydration, calculated on the basis of the dough’s final weight, was maintained at 31%, accounting for the water present in the sourdough. Doughs were prepared with a Kenwood Cooking Chef kitchen machine (Kenwood Limited, New Lane, Havant, UK) using the dough hook accessory and mixed for 12 min at low speed (speed 2) to ensure proper semolina hydration. A piece of dough (200 g) was placed into a graduated glass cylinder, squeezed slightly, and incubated at 28 °C for 7 h. The volume increase was measured every hour. Data were modeled as volume index (VI) = Va − Vc, where Va was the volume of the dough after leavening and Vc was the initial volume. All analyses were performed in duplicate.

The remaining dough was inserted into sealed glass bottles (90 g in each bottle), incubated in a water bath at 28 °C, and the CO_2_ was recorded using an ANKOM gas production system (ANKOM Technology, Macedon, NY, USA). Analyses were performed in duplicate. The instrument measured the gas pressure, and the data were recorded every 15 min up to 7 h using ANKOM GPMx software (version 15.0). Finally, the pressure measured during the test was converted to moles of gas produced using the “ideal” gas law:n = p (V/RT)
where n = quantity of gas in moles (mol), p = pressure in kilopascals (1 psi = 6.8948 kPa), V = headspace volume in the glass bottle (L), R = gas constant (8.3144 L·kPa·K^−1^·mol^−1^), and T = temperature (°K). This was converted into milliliters of gas produced using Avogadro’s law, where 1 mol of gas will occupy 22.4 L at a standard temperature (273.15 °K) and pressure (101.325 kPa). Therefore, the gas measured in moles was converted to gas volume (mL) using the following formula:Gas volume (mL) = n × 22.4 × 1000

The values are expressed as milliliters of CO_2_/90 g dough.

### 2.5. Baking Trials

The performance of S2 (sourdough made with bread residue and wheat middlings) was tested in a bakery trial at the Sunalle artisan bakery in Fonni, Sardinia. Baker’s yeast bread, produced daily at the bakery, was used as the control. The trial was replicated one month later. The sourdough was prepared and inoculated with the selected strains at the Porto Conte Ricerche laboratory, and the first refreshment was made. It was then transferred to the bread factory, where it was refreshed every two days and used for bread-making after 10 days. Sourdough bread was prepared by mixing remilled semolina (100 kg) provided by Brundu S.r.l. (Torralba, Italy), 20 kg of sourdough, 130 g of baker’s yeast (Mistral, Zeus Iba S.r.l., Firenze, Italy), 34.9 L of tap water, and 1.7 kg of sea salt. Dough temperature after mixing was 28 °C. Bulk fermentation was performed at room temperature for 180 min. Control bread was prepared by mixing 100 kg of remilled semolina (Brundu S.r.l. Torralba, Italy), 47 L of tap water, 1.7 kg of sea salt, and 1.5 kg of baker’s yeast (Mistral, Zeus Iba S.r.l., Firenze, Italy). The dough temperature after mixing was 28 °C, and bulk fermentation was performed at room temperature for 40 min. All doughs were prepared by mixing for 25 min in a fork kneader (Fornella RT 160, Sancassiano S.p.a., Roddi d’Alba, Italy). After bulk fermentation, the dough was placed on a mechanical dough sheeter and shaped into sheets of 36 cm diameter and 1 mm thick, then dough sheets were laid onto a conveyor belt and driven within a fermentation chamber set to 32 °C and 90% relative humidity for the final proofing. Sourdough-leavened and yeast-leavened dough sheets stayed in the fermentation chamber for 50 min and 32 min, respectively. Thereafter the dough sheets were conveyed to an electric tunnel oven for the first baking (550 °C for 5 s), where they rapidly puffed up due to the rapid vaporization of their internal moisture. Once baked, the puffed bread was split into two layers and stored in a plastic bag at low temperature (5 °C). After 24 h, the bread layers were toasted in the electric tunnel oven (400 °C for 5 s). After cooling, the bread was packaged using a polyolefin shrink film (Cryovac^®^ CT-304E, Sealed Air Corporation, Elmwood Park, NJ, USA) and brought to the laboratory for analysis.

### 2.6. Rheological Properties of Bread Doughs

Small dynamic oscillatory tests were performed on the doughs at the end of mixing and after sheeting and leavening using a HaakeTM MarsTM iQ rotational rheometer (ThermoFisher Scientific, Waltham, MA, USA) equipped with a serrated parallel-plate geometry (25-mm diameter, 2-mm gap). Strain sweep tests were carried out to identify the linear viscoelastic regions where the moduli were independent of the strain and to choose the target strain to apply to the dough (0.07%). Frequency sweep tests ranging from 0.1 to 10 Hz were then used to further characterize the rheological properties. The storage modulus (G′), the loss modulus (G″), and tan δ (G″/G′) were determined at 10 °C, with three replicates for each sample.

### 2.7. Physicochemical Properties of Carasau Bread

The texture of carasau bread was assessed using a large static deformation test on a TA-XTplus texture analyzer equipped with a 5 kg plugged load cell. Measurement of force (newtons, N) versus time (seconds, s) was recorded using a P/0.5S spherical stainless probe and an HDP/90 heavy-duty platform equipped with a holed plate. The test speed settings were pre-test 5 mm/s, test 0.5 mm/s, and post-test 10 mm/s. Distance was set at 10 mm and trigger force at 0.05 N. Exponent Stable Micro Systems software (v6.1.16) was used for data acquisition and processing. The highest peak in the curve was taken as an indicator of hardness and the number of peaks as an indicator of crunchiness. Twenty sheets of bread were analyzed for each sample.

Carbohydrates, organic acids, and ethanol in carasau bread were determined as for the sourdough (see Section 2.2).

### 2.8. Sensory Properties of Carasau Bread

The analysis was performed on sourdough bread and baker’s yeast bread within a week of being baked. Consumers were screened before participating and disqualified if they had gluten intolerance. Consumers were volunteers, and all received written and oral information about the test and the contents of the assessed products, including allergens. They all gave their informed consent to participate. Information was not used to identify any individual participant, in accordance with a regulation of the European Parliament [23]. The products tested in this study were confirmed safe for consumption, and participants were given the option to withdraw from the study at any time without justification. Consumers received non-monetary compensation for their participation.

The hedonic survey of consumer acceptability and the “check all that apply” test (CATA) were conducted in the Food Science Laboratory of Porto Conte Ricerche.

A total of 104 subjects were recruited for the consumer acceptability test (64 men and 40 women) aged 32–60 years, all living in northern Sardinia. In terms of educational background, the participants were predominantly highly educated: 43% held a master’s or specialist degree, 35% had a bachelor’s degree, and 17% had a PhD or other advanced qualification. Only 5% of respondents had completed their education at the secondary level. Regarding occupation, the largest segment of consumers was public employees (42%), followed by researchers (33%), the self-employed (15%), and teachers (10%). For the evaluation, bread samples were served in closed transparent food container at room temperature within an odor-free testing room, and each consumer had water to rinse his mouth between samples. The samples were labeled with a 3-digit random code and presented in a completely randomized order. Each sample was evaluated based on overall liking of appearance, odor, taste/flavor, and texture. The sensory scores of carasau bread (both baker’s yeast and sourdough bread) were evaluated using an acceptance test and 9-point hedonic scale. Consumers assigned scores from 1 to 9 to the samples, ranging from “extremely dislike” to “extremely like” [24].

For the CATA test, 114 subjects (64 women and 50 men) aged 34–64 years were recruited. In order to identify the sensory attributes to be used in the CATA test, two focus group sessions were conducted with ten expert bakery technicians. The questionnaire was thus created using 25 sensory attributes (Appendix A) and administered via the Smart Sensory Solutions platform (www.smartsensorysolutions.com (accessed on 4 March 2025)). Two bread samples (50 g per sample) were provided in a monadic order in odorless plastic containers labeled with three-digit random codes. To account for the order bias of the CATA attributes, the terms in the questionnaire were randomized according to the procedure suggested by Meyners and Castura [25]. Consumers were asked to select the terms they considered appropriate to describe each sample.

### 2.9. Statistical Analyses

All statistical analyses were run using XLStat version 2018.01 (Addinsoft, New York, NY, USA) and Statgraphics Centurion software (version 16.1.11, Statpont Technologies Inc., Warrenton, VA, USA).

One way-ANOVA with two replications was performed and means were separated by Tukey’s test (*p* < 0.05).

## 3. Results

### 3.1. Sourdough Acidification and Microorganism Growth

The changes in pH and titratable acidity in sourdough samples (S1, S2, S3) before the inoculation (T0) and after daily refreshments (R1–R5) are shown in Figure 1.

At T0, prior to the inoculation of microorganisms, all substrates exhibited a pH of 6.7. In S1, the pH value decreased slightly but significantly (*p* < 0.05) after R1, remaining almost constant throughout the experiment and finishing at pH 6.0 at R5. Concurrently, a slight but significant increase (*p* < 0.05) in TTA values was observed during the refreshments, rising from 2 at T0 to 3.4 after R5, which confirmed its lower overall acidity compared to the other samples. On the other hand, both S2 and S3 sourdoughs showed a significant pH drop (*p* < 0.05) after R1, progressively and significantly (*p* < 0.05) declining to R5 and reaching 4.35 and 4.55 for S2 and S3, respectively. The increased pH observed in S3 was probably due to the higher buffering capacity of semolina compared to breadcrumb. For S2, TTA values progressively increased from 1.75 at T0 to approximately 7.0 at R3, subsequently stabilizing at 7.3 at R5 refreshment. In S3 sample, a rapid increase in TTA values was observed after the first refreshment (from 1.65 at T0 to 5.0), reaching 7.5 at R2 and remaining constant through R5. This finding indicated that incorporating wheat middlings into breadcrumb in S2 made this substrate’s acidification capacity similar to S3 (semolina).

Figure 2 shows the cell density of the presumptive *L. plantarum*, *S. cerevisiae*, and *W. anomalus* as the log_10_ number of colony-forming units (CFU) per gram of sourdough. This was determined after each refreshment step (R1–R5). The cell density after inoculation (T0) was approximately 7.6 log_10_ CFU g^−1^ for *L. plantarum*, 6.3 log_10_ CFU g^−1^ for *S. cerevisiae*, and 7.0 log_10_ CFU g^−1^ for *W*. *anomalus*.

As depicted in Figure 2A, the *L. plantarum* cell density of S1 remained constant throughout the experiment and was significantly lower than in S2 and S3. On the contrary, both S2 and S3 exhibited a significant and rapid increase in *L. plantarum* counts (about 1.5 log cycles) after the first refreshment (R1), reaching levels of 9.0 log_10_ CFU g^−1^. They generally maintained these high levels throughout the subsequent refreshments (R2–R5), suggesting a robust fermentation capacity that aligned with the rapid acidification previously observed.

As illustrated in Figure 2B, the cell counts of presumptive *S. cerevisiae* in the different substrates showed the same trends as *L. plantarum*. The cell density of S1 remained statistically similar to its T0 value and was significantly lower than that of S2 and S3 after all refreshment steps. Conversely, both S2 and S3 exhibited a statistically significant increase in cell counts after cycle R1 and then consistently maintained high cell counts, ranging from approximately 7.2 to 7.8 log_10_ CFU g^−1^ from R2 to R5. A significant difference between them was found only at R4 and R5, where S2 performed better than S3.

Figure 2C, detailing the growth of presumptive *W. anomalus*, shows that cell counts of S1 remained similar to its T0 value (7.0 log_10_ CFU g^−1^) and were significantly lower than S2 and S3 throughout all refreshment steps, as previously observed for *L. plantarum* and *S. cerevisiae*. In contrast, S2 and S3 demonstrated a similar and significant increase in cell counts after the R1 cycle (around 7.7 log_10_ CFU g^−1^) and maintained high levels from R2 to R5.

### 3.2. Sourdough Metabolites

Figure 3, Figure 4 and Figure 5 show the content of lactic acid, acetic acid, glucose, maltose, fructose, and ethanol in sourdoughs. Lactic acid production (Figure 3A) was nearly absent in S1, correlating with the previously observed limited growth of lactic acid bacteria. In contrast, it was detected in both substrates S2 and S3, with higher amounts in S3 (maximum level 0.6%, *w*/*w*) compared to S2 (maximum level 0.5%, *w*/*w*). Remarkably, acetic acid was found in S3 before fermentation (Figure 3B) at a concentration of 0.17% (*w*/*w*), originating from semolina, and persisted throughout the refreshments. In contrast, it was only present in trace amounts in S1 and S2. The different levels of lactic and acetic acid between S2 and S3 further support the variation in TTA values reported in Figure 1.

Simple sugars, including glucose, fructose, and maltose, were detected in all three substrates (Figure 4), while sucrose was absent. Glucose (Figure 4A) was present at low levels in the substrates before fermentation (T0), ranging from 0.06% (*w*/*w*) in S1 to 0.12% in S3. After the first refreshment, its concentration increased to 0.2% in both S1 and S3, probably due to the enzymatic activity from microbial and endogenous enzymes. Afterwards, glucose levels decreased and remained below 0.2% as a result of the microbial activity.

Fructose (Figure 4B), initially present at T0 at low concentrations (ranging from 0.05% *w*/*w* in S3 to 0.08% *w*/*w* in S1 and S2), was almost entirely depleted during fermentation in all the substrates, likely due to microbial consumption.

At the initial time point (T0), maltose was detected in all substrates (Figure 4C). S1 and S3 both showed a concentration of 1% *w*/*w*, while S2 contained the highest concentration at 3.2% *w*/*w*. In S1, the maltose content steadily decreased throughout the refreshments. In S3, a slight increase was found after the first refreshment (R1), followed by a gradual decrease until R5. Notably, in S2, the maltose was significantly higher at T0 than in S1 and S3: its concentration increased to 5% at R1 before gradually decreasing to 3.5% during subsequent refreshments.

As shown in Figure 5, the ethanol content at T0 was negligible in S1–S3. By the first refreshment cycle, ethanol production by the yeast activity progressively increased in all samples, with the lowest content in S1 (0.3%, *w*/*w*) compared to S2 and S3 (approximately 0.5%). From R2 to R5, a clear trend of increasing ethanol concentration was evident in all samples. S1 consistently produced ethanol at a slower rate, gradually rising from R1 to approximately 0.7% *w*/*w* at R5. In contrast, S2 and S3 exhibited higher fermentation efficiency throughout the process. Furthermore, S2 showed most efficiency in ethanol production during the later stages, achieving the highest final concentration by R5 (1.6% *w*/*w*), while S3’s production appeared to plateau towards the end, reaching about 1.2% *w*/*w*.

### 3.3. Sourdough Consistency

The data reported in Table 1 detail the consistency and cohesiveness values of the sourdoughs measured before fermentation (T0) and after R4. The table also shows the percentage reduction in values at R4 compared to T0. A decrease in both consistency and cohesiveness values was observed at R4 for S1 and S2. This reduction was significantly higher in S2 than in S1 as a result of the better microbial activity in S2 than in S1. In contrast, the decrease in consistency values was lower in S3 than in the other substrates and there was no variation in cohesiveness. This result could have been related to the presence of gluten in the semolina-based substrate, which guaranteed higher consistency and cohesiveness even after fermentation.

### 3.4. Sourdoughs Leavening Capacity and CO_2_ Production

To assess their suitability for bread-making, the leavening performance of S2 and S3 sourdoughs was evaluated and compared. Based on previous observations, S1 was excluded from further analysis. Figure 6 illustrates the dough volume increase over 7 h, while Figure 7 presents the corresponding CO_2_ production for doughs inoculated with 5%, 10%, and 20% of S2 and S3 sourdoughs.

The 5% inoculation resulted in the lowest volume increase, reaching a maximum of 1.5 mL for S2 and 1.9 mL for S3 after 7 h. Increasing the inoculation to 10% produced a more linear increase in volume for both samples, reaching approximately 3.0 mL by the end of the test (Figure 6).

The best leavening results were obtained when doughs were inoculated with 20% sourdough for both samples. The most rapid leavening occurred within the first 6 h, reaching maximum dough volumes of 2.7 mL for S2 and 3.6 mL for S3 (Figure 6).

Concurrently, as reported in Figure 7, S2 and S3 exhibited relatively similar patterns of CO_2_ production across all inoculation percentages. The CO_2_ level with 20% sourdough inoculation was significantly greater than with 10% or 5% inoculation, with the amount continuing to increase up to 7 h. Interestingly, the leavening performance of S2 was nearly comparable to S3, even though, as expected, S3 doughs achieved a final CO_2_ volume higher than S2 (85 mL for S2 and 108 mL for S3 at 7 h).

The volume of doughs inoculated at 20% stopped increasing after the sixth hour in both S2 and S3 (Figure 6), although CO_2_ production kept rising until the seventh hour (Figure 7). This behavior could be attributed to the ethanol produced by fermenting yeast, which may have inhibited dough expansion by acting on gluten extensibility and increasing dough stiffness, as observed by Jayram et al. [26].

### 3.5. Baking Trials

Sourdough bread was made using semolina and a 20% inoculation of S2 sourdough starter. The bakery’s daily production of baker’s yeast bread was used as a control sample. Moisture, pH, and TTA of both doughs and breads, along with the weight of dough sheets after leavening, are reported in Appendix A. Sourdough activity was highlighted by the significant differences in pH and TTA values observed: sourdough bread exhibited pH 5.69 and TTA 7.26 compared to pH 6.25 and TTA 4.64 for the baker’s yeast bread. Furthermore, 99% of the baker’s yeast bread dough sheets had fully puffed up after the first bake, whereas 83% of the sourdough bread dough sheets achieved the same.

### 3.6. Rheological Properties of Bread Doughs

Data of viscoelastic moduli G′, G″, and loss tangent tan δ of dough after mixing and after sheeting and leavening measured at a frequency of 1 Hz are shown in Table 2. Mechanical spectra of G′, G″, and tan δ are provided in the Appendix A. Baker’s yeast samples showed higher G′, G″, and G* values than sourdough samples in both dough and sheets. Mechanical spectra indicated that viscoelastic moduli G′ and G″ increased with oscillatory frequency from 0 to 10 Hz. In contrast, tan δ exhibited complex behavior: it was highest at the lowest frequencies, decreased to a minimum as frequencies increased, and then started to increase. This indicates that at lower frequencies, G′ increased more rapidly than G″ due to the elastic response of the gluten–starch polymer network of the dough. After reaching a minimum, the moduli behavior changed, with G″ increasing more rapidly than G′, indicating an increase in viscous behavior. Therefore, we can argue that the gluten–starch network structure was more elastic at lower frequencies and that it started to lose its strength or break down as frequencies increased. As expected in viscoelastic solids, like dough under low strain, G′ was higher than G″ in both control and S2 for both dough and sheets (Table 2), Interestingly, the S2 dough sheet displayed higher G′ and lower tan δ than the control dough sheet, thus showing a more elastic and strong gluten–starch network structure.

### 3.7. Metabolites of Dough and Carasau Bread

Table 3 shows the amount of organic acids, sugars, and ethanol found in dough after mixing in leavened dough sheets, in bread and in semolina used for bread-making. As expected, lactic acid was found in sourdough bread (0.4%, *w*/*w*), but not in control bread, due to LAB metabolism. On the contrary, acetic acid was detected at higher levels in the control bread than in the sourdough bread, with the highest amount (0.95%, *w*/*w*) observed in leavened sheets. Undoubtedly, the high values of acetic acid found in bread doughs were determined by the high level of acetic acid found in semolina (0.47%, *w*/*w*), as reported in Table 3. Ethanol was present at higher levels in dough prepared with baker’s yeast than with sourdough, as expected, but it disappeared in both breads. High levels of simple sugars were found in both samples. Glucose concentration was lower in S2 dough than in baker’s yeast dough and decreased in both bread samples, most probably due to its involvement in the Maillard reaction. In contrast, fructose and maltose were found at high levels in both breads.

### 3.8. Textural Properties and Sensory Analysis of Carasau Bread

Table 4 shows the textural properties of breads. The highest peak was used as an indicator of hardness, while the number of peaks over a set threshold indicated the crispness. Interestingly, the sourdough bread exhibited significantly lower hardness and slightly higher crispness compared to the control.

The results of the consumer test highlighted a significant difference (*p* < 0.05) in the mean acceptability score, with sourdough bread scoring 7.0 compared to 6.0 for baker’s yeast bread. The box plot of overall liking, reported in Figure 8, showed a lower dispersion of data for sourdough bread than for baker’s yeast bread. This indicated greater agreement among consumers regarding the acceptability of the sourdough bread compared to the control bread. Moreover, the acceptability scores for taste and texture indicated higher values for sourdough bread than for control bread. Among the 25 sensory attributes used in the CATA test, 11 showed significant differences (Appendix A). Consumers chose the attributes tasty, very crunchy, and bran smell for sourdough bread more frequently than for baker’s yeast bread. Furthermore, sourdough bread was found to be very toasted, very rough, and to have a rustic appearance.

## 4. Discussion

Bread accounts for a significant proportion of the world’s daily food waste, and its reuse in the food industry would meet the requirements of the circular economy. The recycling of bread waste has been extensively considered in the scientific literature [4,27]. Due to its high carbohydrate content, bread waste is considered an excellent fermentation feedstock, and methods for producing biochemicals through microbial fermentation have been widely reviewed [1,7]. Most of these studies report on the use of enzymes in order to hydrolyze starch or proteins, thereby improving microbial metabolism. The idea of using untreated bread waste to make fresh bread has also been considered [28,29]. However, this has been found to result in a reduction in bread quality, probably because both gluten and starch are no longer in their native state after the thermal treatment, which causes the gluten protein to denature and starch to gelatinize, followed by retrogradation.

In this study, microbial strains exhibiting proteolytic and amylolytic activity were used to ferment bread residue and prepare a sourdough for making carasau bread. Thus, three substrates were used for sourdough preparation: bread residue (S1), bread residue supplemented with wheat middlings (S2), and semolina (S3) as control.

The substrate consisting only of bread residue (S1) was found to be unsuitable for fermentation. Despite daily refreshment, the cell density of microbial strains did not increase throughout the week (Figure 2). Consequently, the production of lactic and acetic acids was almost undetectable (Figure 3), while sugars were still being consumed (Figure 4). The primary function of sugars in S1 was likely to be the replication of microbial cells, which were reduced by half in the refreshment step. Similar results were previously found in bread waste slurry fermented by lactic acid bacteria for 24 h [12]. This slurry showed signs of stunted microbial development and a low level of acidity, but fermentation improved after addition of amylase and protease enzymes. Additionally, a reduction in organic acid production was noted when over 25% of old bread was used to produce sourdough [30].

The results obtained from the substrate S2 (Figure 1, Figure 2 and Figure 3) confirmed that adding wheat middlings to the bread residue had a positive effect, allowing good microbial growth, similar to the semolina substrate (S3). The recorded acidity and cell density values were similar to those commonly found in spontaneous and inoculated sourdough fermentation [16,31]. Unless LAB counts were comparable in both S2 and S3 over the week, lactic acid production was lower in S2 than in S3. The yeast cell density in S2 was clearly linked to the amount of ethanol produced (Figure 5), a trend also observed for S3. Furthermore, the ratio of yeast to LAB counts in S2 ranged from 1:100 to 1:10, consistent with the existing literature [15].

Even though bread waste has a similar macro-composition to wheat flour, this does not guarantee a comparable fermentation process. The literature identified the limiting growth factors as low micronutrient content, i.e., phosphate and vitamins and nitrogen sources [10,12]. This is in accordance with our results, since middlings derived from the outer wheat kernel layers were rich in ash and proteins. Specifically, wheat middlings used in our study contained high levels of ash (5.01 g/100 g d.w.), proteins (15.87 g/100 d.w.), and germ remnants, evidenced by their high lipid content (6.10 g/100 g d.w.). Beyond their nutritional benefits, wheat middlings offer a practical advantage: they are inexpensive and straightforward to use, differing from the enzymes and salts typically employed for fermenting bread waste. The positive impact of wheat bran on bread waste fermentation has also been corroborated by Verni et al. [32]. The authors found that lactic acid bacteria produced more γ-aminobutyric acid (GABA) in bread supplemented with 15% or 30% of wheat bran, compared to plain bread or enzymatically treated bread.

The addition of middlings clearly impacted the maltose level detected in S2 (Figure 4C), which was higher than in both S1 and S3. The maltose levels in these substrates were similar. As reported in the literature, the amount of maltose in flour, both before and after fermentation, does not exceed 1 g/100 g [33,34]. Before fermentation (T0), substrate S2 contained ca. 3% maltose, a level significantly higher than in S1 and S3 (Figure 4C). This difference suggests that wheat middlings, with their greater number of endogenous enzymes compared to semolina or bread residue, are responsible for this increase [35]. Maltose level in S2 increased further after the first refreshment (R1), probably due to the activation of endogenous α-amylase by the decrease in pH [36]. Subsequently, it decreased as a consequence of microbial metabolism. As shown in Figure 4A, glucose levels in the sourdough samples remained low throughout the experiment, aligning with findings from previous studies on semolina-based sourdough [33,34]. The determination of sugar consumption by microorganisms is always complicated due to the daily supply of raw materials during refreshments and the continuous release of carbohydrates from starch via enzymatic activity, particularly from endogenous flour enzymes. The presence of glucose and maltose in S2 and S3 undoubtedly prevented the microorganisms from starving and promoted their growth (Figure 4). Since the glucose levels in S1 were similar to those in S2 and S3, it is clear that glucose was not the growth-limiting factor for microorganisms in S1.

We successfully scaled up the positive results obtained from the substrate S2 (bread residue and wheat middlings) by using this sourdough at 20% for carasau bread production at the industrial level. We then compared the S2-based doughs and breads to corresponding baker’s yeast-leavened products, which were prepared daily at the bakery. Rheological data collected at the bakery for doughs after mixing and sheets after leavening (Table 2) showed higher values of viscoelastic moduli G′, G″, and G* for the control samples compared to the sourdough samples. This finding is not completely in agreement with the existing literature. Fadda et al. [37] and Fois et al. [38] reported that sourdough addiction typically reduces the storage modulus G′ due to lower rigidity. Here, the sourdough is made with unconventional raw materials, i.e., bread residue and middlings, which cannot impart the typical viscoelastic properties of a native gluten–starch matrix. Interestingly, G′ and G* were higher in our sourdough after mixing and also in the leavened sheets, where tan δ was lower. This suggests a stiffer and more solid structure in the sourdough gluten–starch leavened network compared to the baker’s yeast sample. Note that the higher G′ and G*, the lower the dough expansion during leavening is expected to be, a finding that aligns with our previous observation.

Sourdough activity in bread was confirmed by pH and TTA values, as well as by the amount of lactic acid present. The higher amount of acetic acid in control bread compared to the sourdough bread could have been a result of the baker’s yeast secondary metabolism [39]. Furthermore, the acetic acid production by the LAB strains was found to be reduced when *L. plantarum* was in association with *S. cerevisiae* [40]. The presence of acetic acid in the semolina used for bread-making undoubtedly determined the high levels of this metabolite in both samples, yeasted and sourdough bread. Since this organic acid is not a natural component of semolina, we hypothesize it was introduced either during wheat harvesting or as a cleaning agent (i.e., vinegar) in the mill.

The sensory analysis results confirmed the consumers’ preference for sourdough bread, as already observed in a previous study [13]. Sourdough bread was perceived as both tastier and crunchier than yeast bread, in accordance with our texture analysis results. The use of sourdough prepared with old bread was already found to improve the bread’s taste and smell compared to simply adding bread waste to the dough [30]. Meral and Karaoglu [29] reported that using up to 15% stale bread in the production of bread did not affect the sensory properties positively or negatively. They observed a decline in overall acceptability beyond this 15% threshold. In contrast, our study found that using bread residue for sourdough fermentation not only maintained but actually improved the quality of the resulting bread.

## 5. Conclusions

This study successfully demonstrated the potential of valorizing carasau bread residue through sourdough fermentation using selected bacteria and yeast strains. This process transforms a waste product into a valuable resource for the production of new bakery products. Carasau bread residue represents 10% of daily production, representing a low cost for bakeries in terms of energy, human resources, and raw materials. The proposed approach not only addresses the growing problem of food waste but also offers a sustainable solution for baking industries, in line with circular economy principles. In particular, carasau residue supplemented with wheat middlings proved to be an effective substrate for the growth of sourdough microorganisms, leading to an improvement in the sensory properties of the final product. Moreover, the absence of contaminant microorganisms and the low moisture content of bread residue, reducing the likelihood of mold contamination, suggested that its use does not represent a health risk.

Implementing this approach into existing production processes can significantly reduce waste and create innovative products with physicochemical properties comparable to commercial carasau bread made with baker’s yeast, but with distinctive sensorial characteristics.

This work opens new prospects for developing sustainable and high-value-added food products, emphasizing the importance of finding creative solutions for managing food waste. Future research could refine the fermentation process and extend the application of this technology to other types of food residues, solidifying the role of sourdough fermentation in fighting food waste and promoting a more sustainable food industry.

## Figures and Tables

**Figure 1 microorganisms-13-01745-f001:**
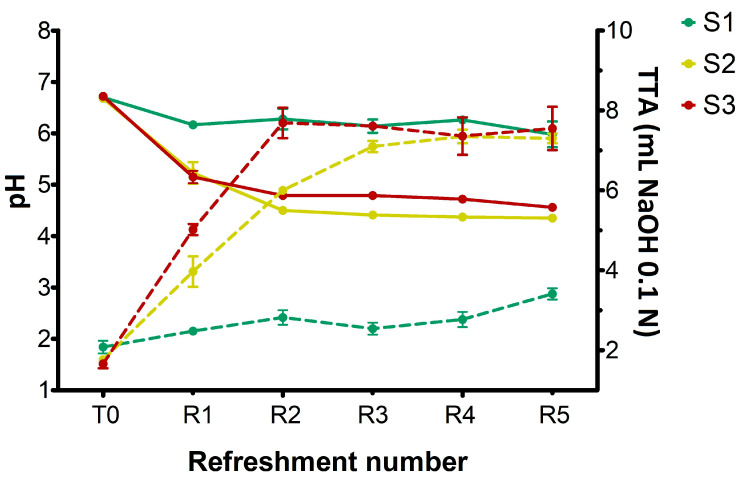
Changes in pH (solid lines) and total titratable acidity (TTA) values (dotted lines) in sourdough samples (S1, S2, S3) after inoculation (T0) and daily refreshments (R1–R5). TTA is reported as mL of NaOH 0.1 N in 10 g of sample. Bars indicate standard deviation.

**Figure 2 microorganisms-13-01745-f002:**
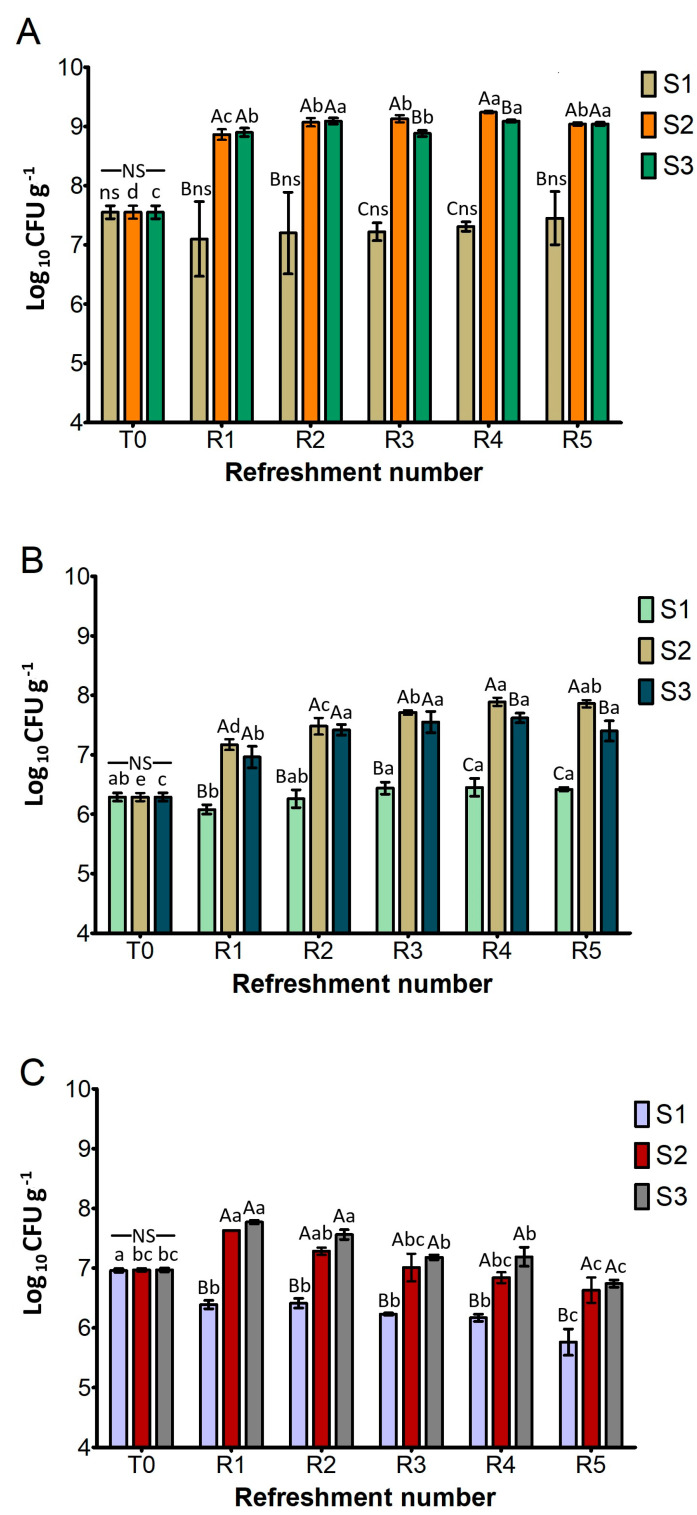
Cell density (log_10_ CFU g^−1^) of presumptive *L. plantarum* (**A**), *S. cerevisiae* (**B**), and *W. anomalus* (**C**) in sourdough samples (S1, S2, S3) after daily refreshment. T0 refers to the cell density after inoculation of microbial starter. Bars indicate standard deviation. Different uppercase letters indicate significant differences among samples within each refreshment time (Tukey’s test at *p* < 0.05). Different lowercase letters indicate significant differences for each sample among refreshments (Tukey’s test at *p* < 0.05). Not significant = ns/NS.

**Figure 3 microorganisms-13-01745-f003:**
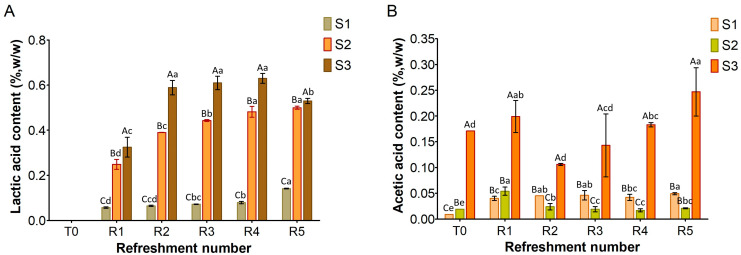
Amount of lactic acid (**A**) and acetic acid (**B**) in sourdough samples (S1, S2, S3) refreshed for 5 days. Bars indicate standard deviation. Different uppercase letters indicate significant differences among samples within each refreshment time (Tukey’s test at *p* < 0.05). Different lowercase letters indicate significant differences for each sample among refreshments (Tukey’s test at *p* < 0.05).

**Figure 4 microorganisms-13-01745-f004:**
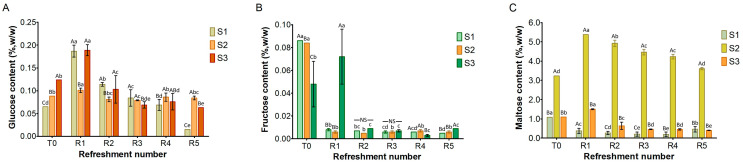
Content of glucose (**A**), fructose (**B**), and maltose (**C**) in sourdough samples (S1, S2, S3) refreshed for 5 days. Bars indicate standard deviation. Different uppercase letters indicate significant differences among samples within each refreshment time (Tukey’s test at *p* < 0.05). Different lowercase letters indicate significant differences for each sample among refreshments (Tukey’s test at *p* < 0.05). Not significant = NS.

**Figure 5 microorganisms-13-01745-f005:**
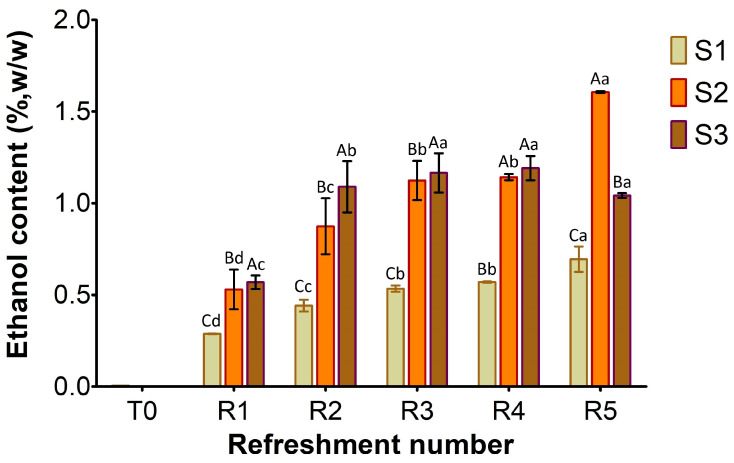
Ethanol concentration in sourdough samples (S1, S2, S3) refreshed for 5 days. Bars indicate standard deviation. Different uppercase letters indicate significant differences among samples within each refreshment time (Tukey’s test at *p* < 0.05). Different lowercase letters indicate significant differences for each sample among refreshments (Tukey’s test at *p* < 0.05).

**Figure 6 microorganisms-13-01745-f006:**
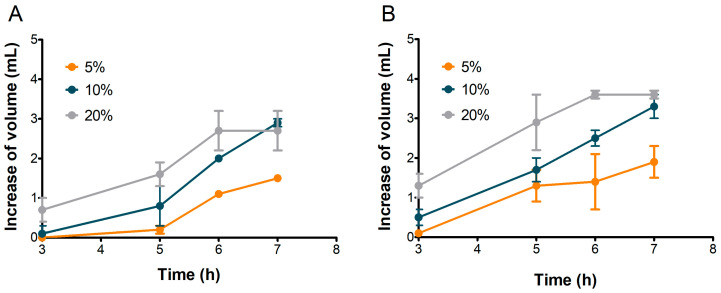
Increase in volume in doughs inoculated with 5%, 10%, and 20% of S2 (**A**) and S3 (**B**) sourdoughs. Values are in mL, calculated as differences between values at each time and at time zero. Bars indicate standard deviation.

**Figure 7 microorganisms-13-01745-f007:**
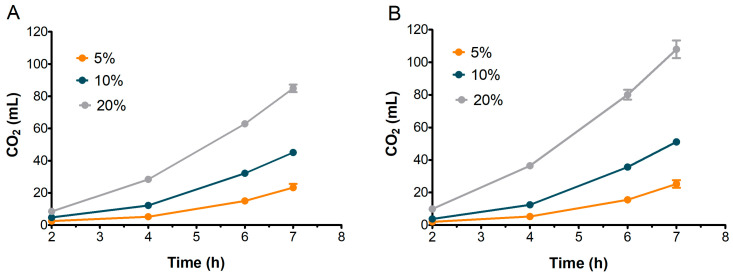
Evolution of CO_2_ production in doughs inoculated with different percentages of S2 (**A**) and S3 (**B**). Bars indicate standard deviation.

**Figure 8 microorganisms-13-01745-f008:**
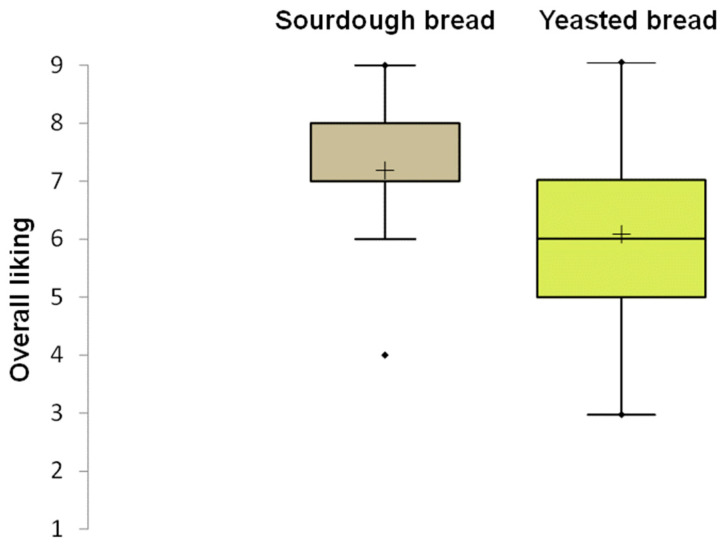
Box plot showing the distribution of liking scores for carasau bread samples. The median (horizontal line) and the mean value (+) are highlighted within each box. The box limits indicate the range of the central 50% of the data. The whiskers indicate minimum and maximum values. Outliers are represented as points (

) beyond the whiskers.

**Table 1 microorganisms-13-01745-t001:** Consistency and cohesiveness values in sourdough samples before fermentation (T0) and after the fourth refreshment (R4). The percentage reduction in values at R4 is reported. Lowercase letters indicate significant differences (*p* < 0.05). Standard deviations are reported.

Substrate	Time	Consistency		Cohesiveness	
		(N s)	Reduction	(N)	Reduction
S1	T0	31.7 _a_ ± 1.06		−2.12 _a_ ± 0.04	
	R4	10.70 _b_ ± 0.47	66%	−1.24 _b_ ± 0.05	40%
S2	T0	21.42 _a_ ± 1.14		−1.36 _a_ ± 0.07	
	R4	3.56 _b_ ± 0.03	83%	−0.18 _b_ ± 0.00	87%
S3	T0	6.04 _a_ ± 0.04		−0.30 _a_ ± 0.01	
	R4	5.17 _b_ ± 0.30	14%	−0.30 _a_ ± 0.03	0%

**Table 2 microorganisms-13-01745-t002:** Dynamic oscillatory data recovered at a frequency of 1 HZ in doughs at the end of mixing and in dough sheets after leavening for the baker’s yeast-leavened dough and the dough leavened by sourdough made with S2. Lowercase letters indicate significant differences (*p* < 0.05).

	G′	G″	tan δ
Samples	(Pa)	(Pa)	
Bakers’ yeast dough	98,253 _bc_	35,120 _b_	0.36 _bc_
Bakers’ yeast sheets	84,641 _c_	39,454 _b_	0.47 _a_
S2 dough	145,333 _a_	50,495 _a_	0.35 _c_
S2 sheets	108,376 _b_	40,697 _b_	0.37 _b_

**Table 3 microorganisms-13-01745-t003:** Organic acids, sugars, and ethanol in doughs, sheets, and breads made using baker’s yeast or S2 sourdough. Lowercase letters indicate significant differences in column (*p* < 0.05).

Samples	Lactic Acid (% *w*/*w*)	Acetic Acid (% *w*/*w*)	Glucose (% *w*/*w*)	Fructose (% *w*/*w*)	Maltose (% *w*/*w*)	Ethanol (% *w*/*w*)
Baker’s yeast dough	0.00 _d_ ± 0.00	0.68 _b_ ± 0.01	0.94 _b_ ± 0.20	1.02 _a_ ± 0.01	3.45 _c_ ± 0.05	0.31 _b_ ± 0.03
Baker’s yeast sheets	0.03 _c_ ± 0.04	0.95 _a_ ± 0.14	1.30 _a_ ± 0.08	0.42 _d_ ± 0.02	2.44 _d_ ± 0.05	0.52 _a_ ± 0.01
Baker’s yeast bread	0.00 _d_ ± 0.00	0.04 _e_ ± 0.01	0.16 _d_ ± 0.01	0.61 _d_ ± 0.03	3.30 _c_ ± 0.09	0.08 _d_ ± 0.00
S2 dough	0.10 _b_ ± 0.00	0.47 _d_ ± 0.19	0.81 _bc_ ± 0.00	0.78 _b_ ± 0.01	4.22 _b_ ± 0.04	0.18 _c_ ± 0.09
S2 sheets	0.39 _a_ ± 0.23	0.59 _c_ ± 0.15	0.70 _c_ ± 0.32	0.75 _bc_ ± 0.08	4.85 _a_ ± 0.30	0.18 _c_ ± 0.01
S2 bread	0.40 _a_ ± 0.23	0.06 _e_ ± 0.00	0.27 _d_ ± 0.09	0.70 _c_ ± 0.09	4.54 _ab_ ± 0.05	0.04 _e_ ± 0.01
Semolina	0.000	0.47	0.39	0.15	3.73	0.00

**Table 4 microorganisms-13-01745-t004:** Textural properties of carasau bread. Lowercase letters indicate significant differences (*p* < 0.05).

Sample	Highest Peak (N)	Number of Peaks
Baker’s yeast bread	4.34 _a_	7.9 _b_
S2 bread	3.99 _b_	9.3 _a_

## Data Availability

The original contributions presented in this study are included in the article/Appendix A. Further inquiries can be directed to the corresponding author.

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
