# Peer review of "Valorizing *Carasau* Bread Residue Through Sourdough Fermentation: From Bread Waste to Bread Taste"

_microorganisms, 2025, doi:10.3390/microorganisms13081745_

Round 1
Reviewer 1 Report
Comments and Suggestions for Authors
The paper “Valorizing Carasau Bread Residues through Sourdough Fermentation: from Bread Waste to Bread Taste” reports the recycle of surplus bread into new bread by transformation of bread residues supplemented with wheat bran into new sourdough via fermentation with a mixture of lactic acid bacteria and yeasts promoting the circular economy for the Carasau bread bioprocess. The objective was clearly specified in the abstract and at the end of the introduction section, where it was mentioned that several studies have dealt with the use of bread waste as a food ingredient and as a feedstock for microbial fermentation. Therefore, the present study lacks originality, and there are no novel technological advances in this work.
The manuscript shows a good understanding of technical terms, writing style, use of practical methodology, and statistical analysis, including the sensory analysis for the newly produced bread. However, in my opinion, the analysis and discussion of the relationship of the acidification and microorganisms’ growth with the metabolomic data are too weak, and the use of the principal component analysis (PCA) technique to evaluate the differences in fermentation parameters among the fermentation products seems like it was not properly applied or justified. For example, for the correct use of the PCA technique, the measurement of multiple parameters should be performed simultaneously in the same experimental unit to find critical information and reduce dimensionality in a dataset, but in the present work, different experimental units were used for each substrate mixture. Also, for the values of the correlation coefficients to be reliable, it is recommended that each variable should contain at least 10 observations, but in the present work, it was not clear the number of repetitions were used, and the correlation matrix was not included. Nevertheless, after the PCA, the results were well presented and discussed. Therefore, I suggest removing the PCA section and improving the analysis and discussion of the relationship of the acidification and microorganisms’ growth with the metabolomic data.
In consequence, I must conclude that this manuscript cannot be published under the current status, but after the recommendations have been attended.
General comments
The background information is acceptable; however better understanding of the metabolomic relationship with the microorganism’s species should be stressed.
Author Response
Thank you very much for taking the time to review this manuscript. The Authors greatly appreciate all the positive and constructive comments from the Reviewers and their suggestions to improve the quality of our manuscript.
The manuscript has been thoroughly revised in line with the reviewers’ suggestions. Accordingly, we have made changes, corrections, clarifications, and performed new calculations to answer to all the concerns. Figures and Table have been reviewed, including the supplementary material.
We hope you find the corrections useful and approve of them.
All the manuscript changes are highlighted in red and in track-changes in the re-submitted files.
Please find below the point-by-point responses to the Reviewer comments, highlighted in red
Reviewer 1
Comments and Suggestions for Authors
The paper “Valorizing Carasau Bread Residues through Sourdough Fermentation: from Bread Waste to Bread Taste” reports the recycle of surplus bread into new bread by transformation of bread residues supplemented with wheat bran into new sourdough via fermentation with a mixture of lactic acid bacteria and yeasts promoting the circular economy for the Carasau bread bioprocess. The objective was clearly specified in the abstract and at the end of the introduction section, where it was mentioned that several studies have dealt with the use of bread waste as a food ingredient and as a feedstock for microbial fermentation. Therefore, the present study lacks originality, and there are no novel technological advances in this work.
The manuscript shows a good understanding of technical terms, writing style, use of practical methodology, and statistical analysis, including the sensory analysis for the newly produced bread. However, in my opinion, the analysis and discussion of the relationship of the acidification and microorganisms’ growth with the metabolomic data are too weak, and the use of the principal component analysis (PCA) technique to evaluate the differences in fermentation parameters among the fermentation products seems like it was not properly applied or justified. For example, for the correct use of the PCA technique, the measurement of multiple parameters should be performed simultaneously in the same experimental unit to find critical information and reduce dimensionality in a dataset, but in the present work, different experimental units were used for each substrate mixture. Also, for the values of the correlation coefficients to be reliable, it is recommended that each variable should contain at least 10 observations, but in the present work, it was not clear the number of repetitions were used, and the correlation matrix was not included. Nevertheless, after the PCA, the results were well presented and discussed. Therefore, I suggest removing the PCA section and improving the analysis and discussion of the relationship of the acidification and microorganisms’ growth with the metabolomic data.
In consequence, I must conclude that this manuscript cannot be published under the current status, but after the recommendations have been attended.
General comments
The background information is acceptable; however better understanding of the metabolomic relationship with the microorganism’s species should be stressed.
Response: We want to thank the reviewer for the critical comments. We believe that this scientific article contributes to the existing body of knowledge by proposing a new method of recycling bread waste. This method could help the bakers to save money, while improving the bread quality through sourdough fermentation.
As suggested by the reviewer, the PCA section has been removed from the manuscript, and further speculation has been added in the discussion about the relationships between microorganisms and metabolomics data.

Reviewer 2 Report
Comments and Suggestions for Authors
The written below is a detailed peer review of the manuscript "Valorizing Carasau Bread Residues through Sourdough Fermentation: from Bread Waste to Bread Taste". Hence, the proposed approach not only addresses the growing problem of food waste but also offers a sustainable solution for the baking industry, which is in line with circular economic principles. In particular, Carasau residues supplemented with wheat middlings proved to be an effective substrate for the growth of sourdough microorganisms, leading to an improvement in the sensory properties of the final product.
However, the manuscript addresses an important topic—valorization of bread waste in the context of circular economy, focusing on Carasau bread residues. The experimental approach is straightforward, and the manuscript is generally well-organized. However, several areas require clarification, additional data, or improved presentation to meet the standards of a high-ranked journal.
Major Comments
- Abstract – Clarity and Quantitative Data
Lines 11-26: The abstract summarizes the work well, but lacks quantitative results. Please include key data (e.g., improvement percentages, microbial counts, sensory scores) to strengthen the impact.
- Introduction – Bread Waste Data
Lines 30-41: The introduction provides a strong rationale, but the cited bread waste figures (900,000 tons/year) lack a reference to the specific country or region. Please clarify the scope.
- Introduction – Literature Context
Lines 43-70: The literature review is comprehensive but would benefit from a more apparent distinction between previous studies using enzymes/salts and your approach using wheat middlings.
- Introduction – Study Rationale
Lines 78-89: The rationale for choosing wheat middlings as a supplement is not sufficiently justified. Please elaborate on why middlings were selected over other supplements.
- Materials and Methods – Substrate Ratios
Lines 93-97: The choice of mixing ratios (1:2.5 for S1/S2, 1:1 for S3) is not justified. Explain why these specific ratios were selected and whether they reflect industrial practice.
- Materials and Methods – Drying Conditions
Lines 99-100: Drying at 130°C for 20 min may affect the nutritional and functional properties of bread residues. Please discuss potential impacts and justify the choice.
- Materials and Methods – Microbial Strains
Lines 108-115: The selection criteria for the microbial strains are briefly mentioned. Please provide references for the claimed properties (e.g., amylolytic activity, aroma profile)
- Materials and Methods – Replicates and Statistical Analysis
Lines 154, 247: It is stated that analyses were done in triplicate or duplicate. Please clarify for each experiment and specify the statistical methods used throughout the study. “Some time four replications are written”
- Methods – Control Bread
Lines 98, 87: The use of semolina as a control is appropriate, but it is unclear if a commercial baker’s yeast-only control bread was also included for sensory comparison.
- Methods – Microbial Enumeration
Lines 134-144: The differentiation of yeast species on W.L. agar by color is mentioned. Please provide a reference or supporting data (e.g., images, colony counts) in the supplementary materials.
- Methods – Enzymatic Kits
Lines 162-177: The use of commercial kits is appropriate, but please specify the detection limits, calibration procedures, and validation steps.
- Methods – Sourdough Fermentation Protocol
Lines 125-133: The back slopping protocol is described, but the rationale for five days of refreshments and the choice of fermentation temperature (25±1.5°C) should be justified.
- Results – Microbial Growth Data
Results section (not included in excerpt): Please present microbial growth curves with error bars and statistical analysis for all substrates.
- Results – Physicochemical Properties
Results section: Include a table summarizing key physicochemical parameters (pH, TTA, moisture, protein, fiber, etc.) for all substrates and breads.
- Results – Consumer Acceptance
Lines 317: Consumer acceptance is referenced. Please clarify how many consumers participated, their demographics, and how acceptance was measured.
- Results – Statistical Significance
Throughout: Clearly indicate which differences are statistically significant and provide p-values.
- Discussion – Comparison to Previous Studies
Discussion section: Compare your findings quantitatively to previous studies on bread waste valorization, especially regarding microbial performance and bread quality.
- Economic and Environmental Impact
Discussion section: Discuss the potential economic and environmental benefits of your approach with quantitative estimates, if possible.
- Limitations
Discussion section: Address limitations such as scalability, potential safety concerns, and shelf-life of the new bread.
- Figures and Tables – Quality and Clarity
Throughout: Ensure all figures and tables are clearly labeled, with appropriate legends and statistical annotations.
- Supplementary Material
Lines 142, 115: Referenced figures and data (e.g., Figure S1, sensory analysis data) be sure that table is included in the supplementary material.
- References – Formatting and Completeness
Throughout: Some references lack complete information (e.g., DOI, full journal names). Please check and update all references.
- Language and Style
Throughout: The manuscript is generally well-written but contains some minor grammatical errors and awkward phrasing. A thorough language edit is recommended.
- Title and Keywords
Lines 1-27: The title and keywords are appropriate, but consider adding "circular economy" as a keyword.
Minor Comments
Line 27: Typo in "Lactiplantibacillu plantarum" (should be "Lactiplantibacillus plantarum").
Line 40: Hyphenation issue in "resources- including water, land, energy, and those involved in raw material production, transportation and manufacturing-are also wasted."
Line 148: "According the AACC combustion method" should be "According to the AACC combustion method".
Finally, the manuscript presents a novel and relevant approach to bread waste valorization. However, several methodological details are missing, and key results (predominantly sensory and consumer data) are not sufficiently described or quantified. The discussion would benefit from a more critical comparison with existing literature and a more precise assessment of the practical implications. With substantial revisions and additional data, the paper has the potential to make a significant contribution to the field.
Author Response
Thank you very much for taking the time to review this manuscript. The Authors greatly appreciate all the positive and constructive comments from the Reviewers and their suggestions to improve the quality of our manuscript.
The manuscript has been thoroughly revised in line with the reviewers’ suggestions. Accordingly, we have made changes, corrections, clarifications, and performed new calculations to answer to all the concerns. Figures and Table have been reviewed, including the supplementary material.
We hope you find the corrections useful and approve of them.
All the manuscript changes are highlighted in red and in track-changes in the re-submitted files.
Please find below the point-by-point responses to the Reviewer comments, highlighted in red
Reviewer 2
Comments and Suggestions for Authors
The written below is a detailed peer review of the manuscript "Valorizing Carasau Bread Residues through Sourdough Fermentation: from Bread Waste to Bread Taste". Hence, the proposed approach not only addresses the growing problem of food waste but also offers a sustainable solution for the baking industry, which is in line with circular economic principles. In particular, Carasau residues supplemented with wheat middlings proved to be an effective substrate for the growth of sourdough microorganisms, leading to an improvement in the sensory properties of the final product.
However, the manuscript addresses an important topic—valorization of bread waste in the context of circular economy, focusing on Carasau bread residues. The experimental approach is straightforward, and the manuscript is generally well-organized. However, several areas require clarification, additional data, or improved presentation to meet the standards of a high-ranked journal.
Major CommentsAbstract – Clarity and Quantitative Data
Comments 1: Lines 11-26: The abstract summarizes the work well, but lacks quantitative results. Please include key data (e.g., improvement percentages, microbial counts, sensory scores) to strengthen the impact.
Response 1. The suggested corrections (e.g. pH values and sensory scores) have been included in the abstract.
Introduction – Bread Waste Data
Comments 2: Lines 30-41: The introduction provides a strong rationale, but the cited bread waste figures (900,000 tons/year) lack a reference to the specific country or region. Please clarify the scope.
Response 2: The suggested correction has been done.
Introduction – Literature Context
Comment 3: Lines 43-70: The literature review is comprehensive but would benefit from a more apparent distinction between previous studies using enzymes/salts and your approach using wheat middlings.
Response 3. To emphasize the potential benefits of the proposed approach some sentences have been included in the Introduction and Conclusion sections.
Introduction – Study Rationale
Comment 4: Lines 78-89: The rationale for choosing wheat middlings as a supplement is not sufficiently justified. Please elaborate on why middlings were selected over other supplements.
Response 4: the rationale for selecting wheat middlings has been provided
Materials and Methods – Substrate Ratios
Comment 5: Lines 93-97: The choice of mixing ratios (1:2.5 for S1/S2, 1:1 for S3) is not justified. Explain why these specific ratios were selected and whether they reflect industrial practice.
Response 5: As suggested, we've explained the specific ratios chosen for preparing the sourdough
Materials and Methods – Drying Conditions
Comment 6: Lines 99-100: Drying at 130°C for 20 min may affect the nutritional and functional properties of bread residues. Please discuss potential impacts and justify the choice.
Response 6: Since the bread residue was obtained from bread previously baked at 500°C, it's plausible that removing moisture at 130°C would have a negligible an impact on their nutritional properties.
Materials and Methods – Microbial Strains
Comment 7: Lines 108-115: The selection criteria for the microbial strains are briefly mentioned. Please provide references for the claimed properties (e.g., amylolytic activity, aroma profile).
Response 7: We've updated the text as recommended, providing a more detailed explanation of the microbial strain selection criteria, along with an additional sentence and supporting reference.
Materials and Methods – Replicates and Statistical Analysis
Comment 8: Lines 154, 247: It is stated that analyses were done in triplicate or duplicate. Please clarify for each experiment and specify the statistical methods used throughout the study. “Some time four replications are written”
Response 8: The suggested corrections have been done.
Methods – Control Bread
Comment 9: Lines 98, 87: The use of semolina as a control is appropriate, but it is unclear if a commercial baker’s yeast-only control bread was also included for sensory comparison.
Response 9: The section “2.8 Sensory properties of Carasau bread” has been revised to include more details regarding the specific breads on which the sensory analysis was performed.
Methods – Microbial Enumeration
Comment 10: Lines 134-144: The differentiation of yeast species on W.L. agar by color is mentioned. Please provide a reference or supporting data (e.g., images, colony counts) in the supplementary materials.
Response 10: The Figure S1 available in Supplementary materials shows a plate culture with the yeast colonies, and the differences between them indicated. The figure was already mentioned in the manuscript (line 143).
Methods – Enzymatic Kits
Comment 11: Lines 162-177: The use of commercial kits is appropriate, but please specify the detection limits, calibration procedures, and validation steps.
Response 11: The authors reported a summary of the test principle. Additional details, including assay performance, linearity, measuring range & sensitivity, limit of quantification (LoQ), precision and accuracy, and sensitive range, can be found in the instructions for the commercial kits. Describing these additional details would be too lengthy and unnecessary, as also suggested by Reviewer 3.
Methods – Sourdough Fermentation Protocol
Comment 12: Lines 125-133: The back slopping protocol is described, but the rationale for five days of refreshments and the choice of fermentation temperature (25±1.5°C) should be justified.
Response 12: The choice of five days and 25°C is based on previous experiences. Some additional sentences are now included in the manuscript, as suggested.
Results – Microbial Growth Data
Comment 13: Results section (not included in excerpt): Please present microbial growth curves with error bars and statistical analysis for all substrates.
Response 13: Since the results of the statistical analysis were now introduced in the figure of microbial growth, we opted for histograms to represent microbial growth as they are clearer and more explanatory than curves for this data. The bars in the histograms indicate the standard deviation.
- Results – Physicochemical Properties
Comment 14: Results section: Include a table summarizing key physicochemical parameters (pH, TTA, moisture, protein, fiber, etc.) for all substrates and breads.
Response 14: The physicochemical parameters of bread have been already reported in Table S2. We attempted to create a single table, including all data for raw material (now listed in the M&M section as text) and bread, but the resulting table was unclear and difficult to understand. We prefer to leave these data in the text.
- Results – Consumer Acceptance
Comment 15: Lines 317: Consumer acceptance is referenced. Please clarify how many consumers participated, their demographics, and how acceptance was measured.
Response 15: Some more demographic informations have been included in the text, as suggested. Acceptance was measured using a hedonic scale and assigning a score from 1 to 9, as reported in the text from line 316 to line 318.
- Results – Statistical Significance
Comment 16: Throughout: Clearly indicate which differences are statistically significant and provide p-values.
Response 16: The suggested corrections has been done.
- Discussion – Comparison to Previous Studies
Comment 17: Discussion section: Compare your findings quantitatively to previous studies on bread waste valorization, especially regarding microbial performance and bread quality.
Response 17: While studies directly focusing on sourdough or bread made with bread waste are limited, we have addressed the comparison of our microbiological results from lines 607 to 621. This section specifically details how the microbial performance in our bread waste-containing substrates aligns with or differs from other relevant findings in the literature. Regarding bread quality, our comparison with existing studies can be found from lines 675 to 682. We believe these sections fulfill the request for comparison by contextualizing our results within the current understanding of bread waste valorization.
- Economic and Environmental Impact
Comment 18: Discussion section: Discuss the potential economic and environmental benefits of your approach with quantitative estimates, if possible.
Response 18: We acknowledge the importance of discussing the potential economic and environmental benefits of our approach, ideally with quantitative estimates. While a comprehensive economic analysis falls outside our primary area of expertise and would ideally be conducted by an economist, we have made an effort to address these aspects to the best of our ability within the current discussion, emphasizing that the bread loss during the Carasau production process is estimated to be 10% in the Introduction and Conclusions sections.
We agree that a more in-depth, quantitatively driven economic and environmental assessment couldform the basis of a valuable future study, building upon the foundational work presented here.
- Limitations
Comment 19: Discussion section: Address limitations such as scalability, potential safety concerns, and shelf-life of the new bread.
Response 19: We've added sentences about the potential safety concern to the Discussion section. The shelf life of the Carasau bread was not discussed because new bread had a moisture under 10%, like the control bread, that gives them a shelf life above 1 year.
- Figures and Tables – Quality and Clarity
Comment 20: Throughout: Ensure all figures and tables are clearly labeled, with appropriate legends and statistical annotations.
Response 20: Figures and Tables have been checked and corrected
- Supplementary Material
Comment 21: Lines 142, 115: Referenced figures and data (e.g., Figure S1, sensory analysis data) be sure that table is included in the supplementary material.
Response 21: OK. The suggested corrections has been done..
- References – Formatting and Completeness
Comment 22: Throughout: Some references lack complete information (e.g., DOI, full journal names). Please check and update all references.
Response 22: references have been updated.
- Language and Style
Comment 23: Throughout: The manuscript is generally well-written but contains some minor grammatical errors and awkward phrasing. A thorough language edit is recommended.
Response 23: OK. As suggested, the manuscript has been revised to improve the English language
Comment 24: Lines 1-27: The title and keywords are appropriate, but consider adding "circular economy" as a keyword.
Response 23: OK. The suggested corrections has been done.
Minor Comments
Line 27: Typo in "Lactiplantibacillu plantarum" (should be "Lactiplantibacillus plantarum"). OK. Done.
Line 40: Hyphenation issue in "resources- including water, land, energy, and those involved in raw material production, transportation and manufacturing-are also wasted." OK. Done.
Line 148: "According the AACC combustion method" should be "According to the AACC combustion method". OK. Done.
Finally, the manuscript presents a novel and relevant approach to bread waste valorization. However, several methodological details are missing, and key results (predominantly sensory and consumer data) are not sufficiently described or quantified. The discussion would benefit from a more critical comparison with existing literature and a more precise assessment of the practical implications. With substantial revisions and additional data, the paper has the potential to make a significant contribution to the field.

Reviewer 3 Report
Comments and Suggestions for Authors
This ms. documents the feasibility of recycling waste bread into a desirable bread product. Some details about the bacterium and yeasts used for the recycling process are provided, while much of the ms. focuses on which substrate is the better for making the recycled sourdough and properties of the dough and final product.
Two big picture issues accompany tis research. First, one might suspect that there is already an adequate use for wast bread, such as animal feed. Second, just like with recycled plastic, it will take some effort to get waste bread recycled if virgin starting materials are cheaply available.
Some content-related questions follow.
lines 52-54 I do not understated what the sentence starting with "To prevent a...." means in the introduction. It seems like an out of place statement without further context.
lines 167-211 The level of detail about these assays is unnecessary. If one wanted, a reader could look these up themselves once provided the name of the exact assay. There also may be easier alternatives available to some of the specific assays used here.
line 246 In the United States, this is simply called the ideal gas law.
Figure 1. Legends should stand alone, so it may be better to state total titratable acids here. TTA also needs defined units.
Figure 2. It is not clear what "a", "b" or "c" mean. Theses are not defined in the legend or on the figures.
Figure 6. As a stand alone figure this may not be informative. None of the components are defined in the legend or figure.
Figure 9. What do the variance lines (vertical) mean? If they indicate standard deviations, then the result sets are not significantly different.
line 674 Should this be vinegar?
The ms. needs revision to improve the presentation. Some examples from the first 100 lines are suggested.
line 16 "residue"; singular is preferable to plural.
lines 20, 31, 32, 73 and elsewhere Excessive use of commas, such as: "microbial performance compared to"; "Every year about 900,000".
line 32 ""in many countries bread is considered a major component".
lines 33-35 "The transformation of bread into waste can have several origins, with staling and spoilage the leading causes, both responsible for a reduction of shelf life.".
line 39 "waste when leftover bread is discarded....".
lines 55, 74 and elsewhere Excessive use of "the", such as: "Currently bread waste is used as...."; "shape due to CO2 expansion....".
line 88 "acceptance of the bread were evaluated.".
Comments on the Quality of English Language
Given above. Pardon, I did not initially see this box.
Author Response
Thank you very much for taking the time to review this manuscript. The Authors greatly appreciate all the positive and constructive comments from the Reviewers and their suggestions to improve the quality of our manuscript.
The manuscript has been thoroughly revised in line with the reviewers’ suggestions. Accordingly, we have made changes, corrections, clarifications, and performed new calculations to answer to all the concerns. Figures and Table have been reviewed, including the supplementary material.
We hope you find the corrections useful and approve of them.
All the manuscript changes are highlighted in red and in track-changes in the re-submitted files.
Please find below the point-by-point responses to the Reviewer comments, highlighted in red
Reviewer3
Comments and Suggestions for Authors
This ms. documents the feasibility of recycling waste bread into a desirable bread product. Some details about the bacterium and yeasts used for the recycling process are provided, while much of the ms. focuses on which substrate is the better for making the recycled sourdough and properties of the dough and final product.
Two big picture issues accompany tis research. First, one might suspect that there is already an adequate use for wast bread, such as animal feed. Second, just like with recycled plastic, it will take some effort to get waste bread recycled if virgin starting materials are cheaply available.
Some content-related questions follow.
Comment 1: lines 52-54 I do not understated what the sentence starting with "To prevent a...." means in the introduction. It seems like an out of place statement without further context.
Response 1: The sentence has been rewritten, in order to clarify the content.
Comment 2: lines 167-211 The level of detail about these assays is unnecessary. If one wanted, a reader could look these up themselves once provided the name of the exact assay. There also may be easier alternatives available to some of the specific assays used here.
Response 2: As Reviewer 2 requested the inclusion of further test details (e.g., assay performance, linearity, LoQ, precision), we prefer to present only a summary of enzymatic principle, directing readers to the manufacturer's provided test instructions for additional informations, as you suggested.
Comment 3: line 246 In the United States, this is simply called the ideal gas law.
Response 3: The sentence has been rewritten to better explain how we proceeded with the calculation of the gas volume.
Comment 4: Figure 1. Legends should stand alone, so it may be better to state total titratable acids here. TTA also needs defined units.
Response 4: OK. The legend and the figure have been modified as you suggested
Comment 5: Figure 2. It is not clear what "a", "b" or "c" mean. Theses are not defined in the legend or on the figures.
Response 5: The capital letters A, B, and C indicate the results of each microbial group, they are reported in the legend (Figure 2. Cell density (log10 CFU g-1) of presumptive L. plantarum (A), S. cerevisiae (B) and W. anomalus (C)…). The letters over the bars (capital and lowercase) indicate the significant differences and are indicated in the legend, as you suggested.
Comment 6: Figure 6. As a stand alone figure this may not be informative. None of the components are defined in the legend or figure.
Response 6: Following the suggestions of Reviewer 1 the PCA section has been removed.
Comment 7: Figure 9. What do the variance lines (vertical) mean? If they indicate standard deviations, then the result sets are not significantly different.
Response 7: Vertical lines in a box plot represent minimum and maximum values. The explanation has been included in the figure legend.
Comment 8: line 674 Should this be vinegar?
Response 8: YES, it is! The word has been corrected.
The ms. needs revision to improve the presentation. Some examples from the first 100 lines are suggested.
All the suggested corrections has been done
line 16 "residue"; singular is preferable to plural. OK, done
lines 20, 31, 32, 73 and elsewhere Excessive use of commas, such as: "microbial performance compared to"; "Every year about 900,000". OK, done.
line 32 ""in many countries bread is considered a major component". OK, done.
lines 33-35 "The transformation of bread into waste can have several origins, with staling and spoilage the leading causes, both responsible for a reduction of shelf life.". OK, done.
line 39 "waste when leftover bread is discarded....". OK, done.
lines 55, 74 and elsewhere Excessive use of "the", such as: "Currently bread waste is used as...."; "shape due to CO2 expansion....". OK, done.
line 88 "acceptance of the bread were evaluated.". OK, done.

Round 2
Reviewer 1 Report
Comments and Suggestions for Authors
The revised version of the paper “Valorizing Carasau Bread Residues through Sourdough Fermentation: from Bread Waste to Bread Taste” that reports the recycle of surplus bread into new bread by transformation of bread residues supplemented with wheat bran into new sourdough via fermentation with a mixture of lactic acid bacteria and yeasts promoting the circular economy for the Carasau bread bioprocess, clearly improved the previous one. Although this study still lacks originality and novelty, the possibility that the protocol for recycling the bread waste provides with a technological alternative for the baker industry was now stressed in the introduction and conclusion sections.
This new version maintains a good understanding of technical terms, writing style, and the use of practical methodology, and improves the significance of both statistical and sensory analysis for the newly produced bread.
As suggested, the analysis and discussion of the relationship between acidification and microorganisms’ growth with the metabolomic data were stressed, and the principal component analysis (PCA) section was properly removed.
In consequence, I suggest that this manuscript can be published under the current status.
Author Response
On behalf of my co-authors, I would like to thank you for your constructive criticism and suggestions.
Reviewer 2 Report
Comments and Suggestions for Authors
Thank you for your thorough and constructive responses to our comments and the substantial revision of your manuscript. We appreciate your efforts in addressing each point, including clarifying methods, updating data and figures, and improving the manuscript’s clarity. Indeed, you notably improved the MS file. However, no more inquiries are required.
Author Response

(The authors gave the same response as above.)

Reviewer 3 Report
Comments and Suggestions for Authors
A revision of this ms. was reviewed. The overall quality and presentation of the work, about the reuse of a specific waste bread, was significantly improved.
The microbiology in the ms. is of concern. It is essentially ancillary to the research. For example, the only microbiological content of the abstract is the name of the three species used in this work.
lines 180-221 The detail given regarding measurement of metabolites is excessive. These are well known methods, not novel. FYI, there are other, probably easier, methods available for these metabolites.
Fig.s 2-5 What are the specific, explicit differences, say, between Aa, Ab, Ac and Ad? It is stated in the legend for Fig. 3 that this denotes that the values are significantly different from each other. But is this meaningful/important? I would recommend that the lower case letters be omitted..
Comments on the Quality of English LanguageThe quality of the presentation was greatly improved. Some further suggestions are given below.
lines 65, 66, 154, 177, 308, 318, 336 and 357 Remove the excess comma.
line 99 "replicates of each treatment....".
lines 120-121 "activity, acidifying properties and ability to produce....".
lines 129-130 "Yeast extract, bacteriological peptone and glucose are not capitalized in this context. Similarly, lines 180-218 metabolites (e.g., glucose) are not capitalized.
lines 159, 161, 626 and 627 "ash" is preferable to "ashes".
line 161 "was determined".
line 167 "0.1 mM NaOH.".
line 174 "duplicate".
line 231 "made over time: ".
line 233 "Analysis was performed....".
line 280 "were prepared by mixing....".
line 284 "90% relative humidity....".
line 297 "Scientific".
line 307 "Measurement of force....".
line 433. "Figure 4".
line 681 "yeast bread".
The References should be checked one more time. For example, ref.s 8 and 16 have taxonomic names that should be in italics, and ref. 14 needs correct use of capitals "Polymers".
Author Response
The manuscript has been further edited in accordance with the reviewer comments. We would like to thank you again for the critical remarks, because each request contributes to the improvement of the paper. Please find below the point-by-point responses to the comments, highlighted in red.
Comment: lines 180-221 The detail given regarding measurement of metabolites is excessive. These are well known methods, not novel. FYI, there are other, probably easier, methods available for these metabolites.
Response: OK. As suggested, we have removed the excessive details of the methods.
Comment: Fig.s 2-5 What are the specific, explicit differences, say, between Aa, Ab, Ac and Ad? It is stated in the legend for Fig. 3 that this denotes that the values are significantly different from each other. But is this meaningful/important? I would recommend that the lower case letters be omitted..
Response: Both the lowercase and uppercase letters were derived from a statistical analysis of the data reported in the Figures 2–5. Uppercase letters indicate differences among sourdough samples within each refreshment. Lowercase letters indicate differences for each sourdough sample along refreshments. We believe both are important to highlight the differences and we prefer to leave the letters in all figures. Perhaps the legend was not clear enough, so we have edited the sentences.
The quality of the presentation was greatly improved. Some further suggestions are given below.
lines 65, 66, 154, 177, 308, 318, 336 and 357 Remove the excess comma. Ok. Done.
line 99 "replicates of each treatment....". Ok. Done.
lines 120-121 "activity, acidifying properties and ability to produce....". Ok. Done.
lines 129-130 "Yeast extract, bacteriological peptone and glucose are not capitalized in this context. Similarly, lines 180-218 metabolites (e.g., glucose) are not capitalized. Ok. Done.
lines 159, 161, 626 and 627 "ash" is preferable to "ashes". Ok. Done.
line 161 "was determined". Ok. Done.
line 167 "0.1 mM NaOH.". This has been modified to “0.1N NaOH” to align with the Figure 1.
line 174 "duplicate". Ok. Done.
line 231 "made over time: ". Ok. Done.
line 233 "Analysis was performed....". Ok. Done.
line 280 "were prepared by mixing....". Ok. Done.
line 284 "90% relative humidity....". Ok. Done.
line 297 "Scientific". Ok. Done.
line 307 "Measurement of force....". Ok. Done.
line 433. "Figure 4". Ok. Done.
line 681 "yeast bread". Ok. Done.
Comment: The References should be checked one more time. For example, ref.s 8 and 16 have taxonomic names that should be in italics, and ref. 14 needs correct use of capitals "Polymers".
Response: Ok. All references have been carefully checked.